# Nonsense Suppression Therapy: New Hypothesis for the Treatment of Inherited Bone Marrow Failure Syndromes

**DOI:** 10.3390/ijms21134672

**Published:** 2020-06-30

**Authors:** Valentino Bezzerri, Martina Api, Marisole Allegri, Benedetta Fabrizzi, Seth J. Corey, Marco Cipolli

**Affiliations:** 1Cystic Fibrosis Center, Azienda Ospedaliero Universitaria Ospedali Riuniti, Via Conca 71, 60126 Ancona, Italy; valentino.bezzerri@ospedaliriuniti.marche.it (V.B.); martina.api@ospedaliriuniti.marche.it (M.A.); marisole.allegri@ospedaliriuniti.marche.it (M.A.); benedetta.fabrizzi@ospedaliriuniti.marche.it (B.F.); 2Department of Pediatric Hematology/Oncology and Stem Cell Transplantation, Cleveland Clinic, Cleveland, OH 44195, USA; coreys2@ccf.org; 3Cystic Fibrosis Center, Azienda Ospedaliera Universitaria Integrata, P.le A. Stefani 1, 37126 Verona, Italy

**Keywords:** inherited bone marrow failure syndromes, nonsense suppression therapy, nonsense mediated decay, ataluren

## Abstract

Inherited bone marrow failure syndromes (IBMFS) are a group of cancer-prone genetic diseases characterized by hypocellular bone marrow with impairment in one or more hematopoietic lineages. The pathogenesis of IBMFS involves mutations in several genes which encode for proteins involved in DNA repair, telomere biology and ribosome biogenesis. The classical IBMFS include Shwachman–Diamond syndrome (SDS), Diamond–Blackfan anemia (DBA), Fanconi anemia (FA), dyskeratosis congenita (DC), and severe congenital neutropenia (SCN). IBMFS are associated with high risk of myelodysplastic syndrome (MDS), acute myeloid leukemia (AML), and solid tumors. Unfortunately, no specific pharmacological therapies have been highly effective for IBMFS. Hematopoietic stem cell transplantation provides a cure for aplastic or myeloid neoplastic complications. However, it does not affect the risk of solid tumors. Since approximately 28% of FA, 24% of SCN, 21% of DBA, 20% of SDS, and 17% of DC patients harbor nonsense mutations in the respective IBMFS-related genes, we discuss the use of the nonsense suppression therapy in these diseases. We recently described the beneficial effect of ataluren, a nonsense suppressor drug, in SDS bone marrow hematopoietic cells ex vivo. A similar approach could be therefore designed for treating other IBMFS. In this review we explain in detail the new generation of nonsense suppressor molecules and their mechanistic roles. Furthermore, we will discuss strengths and limitations of these molecules which are emerging from preclinical and clinical studies. Finally we discuss the state-of-the-art of preclinical and clinical therapeutic studies carried out for IBMFS.

## 1. Introduction

Inherited bone marrow failure syndromes (IBMFS) are characterized by peripheral cytopenia(s) with a hypocellular bone marrow andimpairment in one or more hematopoietic lineages. They are also cancer predisposition syndromes. The classical IBMFS are represented by Shwachman–Diamond syndrome (SDS), Diamond–Blackfan anemia (DBA), Fanconi anemia (FA), dyskeratosis congenita (DC), and severe congenital neutropenia (SCN). Importantly, 10–15% of bone marrow aplasia and 30% of pediatric bone marrow failure disorders are caused by IBMFS, with approximately 65 cases per million live births every year [1]. However, emerging data reveal that IBMFS are underdiagnosed because of decreased recognition or because genetic mutation associated with congenital bone marrow failure are detected only after a malignancy has arisen [2].

Clinical management of IBMFS requires multidisciplinary care and surveillance to detect early emergence of malignancies. Early hematopoietic stem cell transplant (HCT) may correct bone marrow failure and prevent the development of myeloid neoplasia, but it does not affect the risk of solid tumors. However, post-HCT complications, such as graft-versus-host disease and immune dysfunction, frequently occur. As an alternative to HCT, androgen administration may be suitable for patients with FA and DC. Corticosteroids are often effective for patients with DBA, especially for those who lack a compatible donor or are ineligible for HCT transplantation. In addition, recombinant granulocyte colony-stimulating factor (G-CSF) is used to ameliorate severe neutropenia in SCN or SDS and prevent recurrent infections. Long-term use of G-CSF in SCN and SDS has been associated with increased risk of myelodysplastic syndrome (MDS) and acute myeloid leukemia (AML) [3]. No therapies to reduce bone marrow failure or cancer risk in IBMFS have been developed so far. 

It has been estimated that approximately 12% of human genetic disorders are caused by single-nucleotide or out-of-frame nonsense mutations, leading to the generation of premature termination codons (PTC). Correct protein synthesis normally represents the core of biological process for any living organism. For this reason, protein synthesis is regulated at multiple levels and any disruption on each of these steps may cause severe diseases. The therapeutic strategy aimed at overcoming PTC has been defined as “nonsense suppression therapy”. This approach is intended to generate the readthrough of PTC, restoring a full-length protein synthesis. Alternatively, nonsense suppression therapy may be designed to reduce the nonsense mediated decay (NMD) induced by the nonsense mutations. NMD is an evolutionarily conserved defense mechanism of eukaryotic cells that surveys newly synthesized mRNA and degrades transcripts containing a PTC [4,5].

We have tested the efficacy of a small nonsense suppressor molecule, ataluren (PTC124, PTC Therapeutics, NJ) [6], in correcting the basic defect of SDS with promising preclinical results. We are planning a clinical trial of ataluren for SDS. Our preclinical studies might serve as proof of concept for the development of nonsense suppression therapy for other IBMFS. This review will discuss the possible use of nonsense suppression strategy in IBMFS as a novel therapeutic hypothesis.

## 2. Inherited Bone Marrow Failure Syndromes

Bone marrow failure syndromes (BMFS) cluster different disorders characterized by impaired hematopoiesis which lead to selective or global cytopenia. BMFS may be acquired, such as aplastic anemia (AA) and paroxysmal nocturnal hemoglobinuria (PHN), or congenital. IBMFS are classified into those that result in pancytopenia and those limited to deficiency of one or more hematopoietic lineages. FA and DC are characterized by progressive peripheral pancytopenia. Patients with DBA exhibit anemia, whereas SDS and SCN are mostly associated with severe neutropenia. Some of the IBMFS can be characterized by physical anomalies and failure to thrive. IBMFS are all associated with increased risk for the development of MDS/AML and solid tumors.

Approximately 80 different genes have been associated with different IBMFS [7]. Notably, 28% of FA, 24% of SCN, 21% of DBA, 20% of SDS, and 17% of DC mutated alleles show nonsense pathogenic variants (Figure 1). Classes of genes correlate with specific IBMFS, such as genes which encode for proteins involved in DNA repair (FA), telomere biology (DC), and ribosome biogenesis (DBA). Many cancers have been associated with somatic mutations in genes encoding ribosomal proteins [8]. Importantly, defective ribosomal proteins may induce the upregulation of the tumor suppressor gene *TP53* which encode the tumor suppressor protein p53 [9,10]. In this regard, p53 over-activation has been implicated in the pathogenesis of DBA, SDS, and DC [11,12,13,14]. 

### 2.1. Shwachman–Diamond Syndrome

SDS is one of the most common IBMFS, first described in 1964 [16], with an incidence of 1:76,000 [17]. It is characterized by neutropenia, pancreatic exocrine insufficiency, and bone malformations. Almost all patients with SDS exhibit some degree of a hypocellular bone marrow specimen and one fifth of patients develop pancytopenia. Myeloid lineage maturation is severely impaired and neutrophil maturation is blocked at myelocyte-metamyelocyte stage [18]. Mild or severe neutropenia is exhibited by most SDS patients. Thrombocytopenia and anemia are less frequent. SDS patients exhibit a propensity to develop clonal cytogenetic changes in the bone marrow, especially the interstitial deletion of the long arm of chromosome 20, del(20)(q) and chromosome 7 anomalies [19,20,21,22].

About 90% of SDS is caused by mutations affecting the Shwachman–Bodian–Diamond syndrome (*SBDS*) gene, which encodes a protein involved in ribosome biogenesis. Biallelic mutations in *SBDS* have been found in 90% of SDS patients [23,24]. Most importantly, 56% of these patients share the same nonsense mutation c.183-184TA>CT (K62X) in one allele [23]. Nevertheless, the splicing mutation c.258+2T>A, which affects the donor splice site of intron 2, is displayed in the second allele of the almost totality of *SBDS*-mutated patients [23,25]. It turns out that the incidence of nonsense mutations recognized in *SBDS* gene is 20% (Table 1). The apparent inconsistency between the incidence of nonsense mutations in the SDS population (56%) and the allelic incidence of this type of pathogenic variation (20%) is notable because no patients homozygous for the c.183-184TA>CT mutation have been recognized so far. This is probably due to the severity of the homozygous condition which critically impairs the embryonic development. Other genes have been associated with SDS during the last few years, including the *DNAJC21*, Elongation Factor Like GTPase (EFL)-1 and Signal Recognition Particle (SRP)-54 [26,27,28,29]. *DNAJC21*, *EFL1* and *SRP54* are involved in ribosome formation and peptide processing [24]. *DNAJC21* is a co-factor for the pre-60S ribosomal subunit; *EFL1* promotes the release of the eukaryotic initiation factor 6 (eIF6) during ribosome assembly; and *SRP54* supports nascent polypeptide trafficking from the ribosome. Impaired ribosome assembly may lead in turn to reduced number of ribosomes and deficient translation, activating the tumor suppressor protein p53 pathway [9]. Not surprisingly, *TP53* expression is upregulated in SDS [11,12] as well as in other ribosomopathies such as DBA and DC. 

Approximately 15–20% of SDS patients develop MDS or AML, with a progression rate of almost 1% per year [30]. A study conducted on 102 patients from the French SDS registry showed a cumulative risk of MDS/AML of 36% at 30 years of age [31]. Importantly, mutations on *TP53* in patients with SDS have been associated with early leukemogenesis [32]. Overall cancer risk in SDS has been calculated with an observed over expected ratio of 8.5 [9].

### 2.2. Diamond–Blackfan Anemia

Diamond–Blackfan anemia (DBA) is an IBMFS which affects 5 to 10 cases per million newborns in Europe. Almost 55% of DBA patients exhibit mutations in one of 19 different ribosomal protein (RP) genes, resulting in RP haploinsufficiency [53,97]. A plethora of pathogenic variants have been reported to affect genes encoding either for the small ribosomal subunit (RPS) or the large ribosomal subunit (RPL), including *RPS7*, *RPS10*, *RPS15A*, *RPS17*, *RPS19*, *RPS24*, *RPS26*, *RPS27*, *RPS28*, *RPS29*, *RPL5*, *RPL35*, *RPL11*, *RPL15*, *RPL18*, *RPL26*, *RPL27*, *RPL31*, and *RPL35A* [98]. In addition, mutations in *GATA1*, a master transcription factor that is fundamental for normal erythropoiesis [99], and in RPS26 chaperone protein *TSR2* gene [49], have been identified as non-RP mutations in DBA. Of note, *GATA1* mRNA translation deficiency is associated with reduced levels of RPS19, RPL5, and RPL11 proteins [99]. RPS19 was first linked to DBA. *RPS19* mutations account for almost 25% of all DBA cases. Other RP genes most frequently implicated in DBA are *RPL5*, *RPS26,* and *RPL11*. Currently, 130 different variants in *RPS19* have been reported [100]. On the basis of ClinVar database [15], missense mutations are the most common (25%), followed by nonsense mutations, which occur in 20% of *RPS19* mutated-patients (Table 1). In particular, p.Arg94X is the most frequent pathogenic variant recurring in unrelated families carrying *RPS19* mutations [101]. Most nonsense mutations, large deletions and frameshift which generate a PTC, are expected to result in haploinsufficiency [102]. In this regard, it has been suggested that incomplete or aberrant transcripts resulting from PTC located up to 50–55 nucleotides upstream the last exon–exon junction, may be more instable and may induce a rapid transcript turnover through the NMD [36]. 

Although most DBA patients (55–60%) have nofamily history for this disorder, DBA is typically inherited with an autosomal dominant pattern (40–45%) or, less frequently, with an autosomal recessive pattern (10% of patients) [13]. 

Originally, DBA was called “hypoplastic anemia”, since anemia is the main feature of this disease. In fact, anemia is reported in 25% of DBA patients at birth. Furthermore, 95% of DBA patients early develop macrocytic anemia, reticulocytopenia and limited cytopenia. Short stature is frequent among DBA patients, similarly to other IBMFS such as SDS [103]. Other clinical manifestations include cranio-facial abnormalities, including cleft/lip palate (4%), eye malformations (5%), abnormal facies (3%), and microcephaly (2%). Upper-limb malformations have been described, especially regarding thumbs (8%). Urogenital (3%), gonadic (3%), cardiac (3%), and neurological anomalies have been also described together with and developmental issues [104]. Bone marrow biopsies collected from DBA patients show a markedly decrease of erythroid progenitors, which are often sparsely distributed in a few small clusters. Such clusters exhibit delayed or absent maturation of erythroid cells, probably because of excessive tendency to apoptosis or incapability to correctly differentiate [105]. Less frequently, granulopoiesis and megakaryopoiesis result further compromised. Many reports suggest that the limited amount of available ribosomes can specifically impair erythroid differentiation, although the molecular mechanisms that may underlie this deficiency remain poorly clarified. In vivo experiments conducted in zebrafish provided some clues about the link between ribogenesis and anemia. Zebrafish models defective for *RPS19* or *RPL11* displayed decreased globin protein translation in erythroid cells [106]. Ribosome machinery is particularly essential during erythroid differentiation, as the highest rate of protein synthesis occur in bone marrow progenitors undergoing erythroid lineage commitment [107]. 

Additionally, many RPs show extra-ribosomal functions, such as the induction of p53-dependent cell cycle arrest and control of apoptosis. Loss of a RP is often found in the presence of concurrent p53 mutations in several forms of cancer [8]. In fact, likeother IBMFS, DBA patients have a predisposition to develop solid tumors and hematological malignancies. The rate of any solid tumor or leukemia is 5.4-fold higher in DBA patients than in healthy population [108]. According to the Diamond–Blackfan Anemia Registry (DBAR), the risk for a solid tumor including colon adenocarcinoma, osteosarcoma, lung cancer, Wilms tumor, osteogenic sarcoma, and female genital cancer, increasesby at 30 years of age. AML generally occurs only after the fourth decade of life [109].

### 2.3. Fanconi Anemia

Fanconi anemia is an autosomal recessive or X-linked recessive IBMFS caused by mutations in almost 22 genes [110]. FA patients show variable hematological and non-hematological manifestations. FA patients may exhibit short stature (40%) and developmental delay (10%), typical skin pigmentation like “café au lait” spots (<50%), thumbs and skeletal abnormalities (20–30%), abnormal male gonads (25%), microcephaly (20%), eye anomalies (20%), structural renal defects (20%), abnormal ears (10%), and hearing loss (75%). However, 25–40% of patients lack physical abnormalities [104]. At diagnosis, 77% of FA patients exhibit hematological abnormalities, in particular mild-to-moderate cytopenia [111]. A study conducted on 180 Italian patients, from 151 different families, reported that 96% of subjects may develop hematological abnormalities during their disease course and, eventually, 8% may show hematological malignancy, mostly featured by myelodysplastic syndromes (MDS) and acute myeloid leukemia (AML) [112]. In this regard, it has been estimated that FA patients have a 785-fold increased risk of AML evolution from MDS compared to general population [113]. BMF and AML onset in FA begins in childhood [114]. In addition to their hematological condition, patients with FA also have a higher risk of solid cancers. In particular, some evidences suggest that FA patients have an almost 700-fold greater risk of developing carcinomas affecting head, neck and anogenital tissue than the general population by the age of 50 years [115]. According to the Italian RIAF registry, 11% of the Italian FA patients were diagnosed with solid cancers. Although that analysis was limited by the small number of events, large part (44%) of those cancer was represented by head and neck squamous carcinomas, 11% by liver carcinoma, whereas thyroid, breast, and genital tract carcinomas accounted for 7% of solid tumors, each [112]. Similar incidence was observed by the National Cancer Institute’s registry (NCI) prospective longitudinal cohort study, which enrolled 130 FA families [116]. 

FA mutated genes affect Fanconi anemia complementation group (FANC) proteins which are involved in DNA crosslinks repair (Table 1). Accordingly, FA cells exhibit spontaneous and induced chromosome instability, showing chromatid gaps and breaks, interchanges, radial figures, endoreduplication, and chromosome gain or loss [117,118]. Several genes have been associated with FA, including *FANCA*, *FANCC*, *FANCD1*, *FANCD2*, *FANCE*, *FANCF*, *FANCG*, *FANCI*, *FANCJ*, *FANCL*, *FANCN*, *FANCP*, *FANCQ*, *FANCT*, *FANCU*, *FANCV,* and *FANCW* [119]. However, almost 66% of all FA patients harbors mutations in *FANCA* gene, 14% in *FANCC* and 9% in *FANCG* [65]. Other genes are accounting only for 0.1–4% of cases. *FANCB* is the only one located in the X-chromosome. Homozygous mutations in *FANCD1* (also known as *BRCA2*) lead to increased susceptibility to breast, ovarian, and pancreatic cancer [120]. On the basis of ClinVar database [15], 955 genetic variants of *FANCA* have been registered, 301 of whose have been identified as pathogenicor likely-pathogenic. The higher mutation rate observed in FANCA may be explained by *FANCA* gene huge size (43 exons) as well as by the presence of repetitive sequence elements, which may promote unequal recombination and ultimately originate deletions. Thus, the instability of *FANCA* could partially justify why its variations account for about two-thirds of all FA patients [68]. Most *FANCA* mutations are rare, inherited exclusively in few families, with the exception of some founder mutations, as the c.295C>T (Q99X) nonsense mutation, which has been largely spread within the Spanish Gipsy individuals [70]. Recently, an international cohort of 159 *FANCA*-mutated families has been characterized, outlining the extensive heterogeneity of *FANCA* pathogenic variants [65]. Nonsense mutations in *FANCA* account for almost 14%. Similarly, *FANCC* and *FANCG* nonsense mutations account for 33% and 24% of FA pathogenic cause, respectively (Figure 1). Data from the Italian RIAF registry showed that the incidence of nonsense mutations due to SNPs in the Italian cohort (24%) [63] is in line with which we calculated, although deletions remain the most common genetic abnormalities, accounting for 29%.

### 2.4. Dyskeratosis Congenita

Dyskeratosiscongenita (DC) is a cancer-predisposition IBMFS caused by mutations involving genes that regulate telomere maintenance (Table 1). DC is associated with a spectrum of clinical conditions that arise from short and dysfunctional telomeres, termed telomere biology disorders. DC can be inherited via X-linked or autosomal dominant, rarely autosomal recessive, patterns. The rare autosomal recessive conditions are due to variants in *NOP10* and *NHP2* ribonucleoproteins, poly(A)-specific ribonuclease (*PARN*) and repeat containing antisense to *TP53* (*WRAP53 WD*). Most patients exhibit mutations in *DKC1*, which is associated with X-linked inheritance. Conversely, *TERC*, *TERT*, *TINF2*, *RTEL1*, *ACD,* and *CTC1* genes are identified to segregate with an autosomal dominant pattern [121]. A wide range of features are associated with DC, including developmental delay, hematological defects, liver or lung fibrosis, and physical abnormalities [122]. The most common physical alterations are lacrimal duct abnormalities, esophageal and urethral stenosis, early grey hair and eyebrows, osteopenia, and poor dentition [104]. Moreover, 90% of DC patients develop cytopenia by the fourth decade of life in at least one lineage, as well as aplastic anemia and immunodeficiency. Clinical manifestations, age of onset and severity of symptoms depend on which gene is mutated and vary among DC patients due to incomplete penetrance and genetic anticipation phenomenon.

The pathogenesis of DC is incompletely understood. Performing fluorescence in situ hybridization (FISH) experiments, Alter and colleagues have shown that aging is associated with telomere shortening. The reduction of age-adjusted value of telomere length is remarkably associated with hematologic status, providing a quantitative measure of disease severity [123]. Thus, it has been proposed that telomere shortening can lead to stem cell pool exhaustion, especially in tissues characterized by high cellular turnover such as blood and epithelium. The balance between the attrition due to the replication process and telomerase-mediated repeat addition is an important determinant for stem cell self-renewal ability and tissue regeneration. Defects in telomere length may provide an intuitive explanation about the link with carcinogenesis, but the exact mechanism remains poorly clarified. It has been suggested that shortened telomeres cause chromosome instability with subsequent cell senescence due to the activation of p53-dependent signaling pathways [124]. Telomere erosion accumulation among successive generations is responsible for precocious onset of the disease and generally leads to severe phenotypes [125]. Mutations in *TERT* gene, which encodes telomerase reverse transcriptase protein, occur in 1% of DC patients and are associated with mild to severe anemia, liver disease, and often with pulmonary fibrosis. Germline mutations of *TERT* were detected in almost all sample from an adult patients’ cohort suffering from idiopathic pulmonary fibrosis [126]. Interestingly, large deletions or nonsense mutations resulting in total loss of functional proteins are rarely identified in DC patients. According to ClinVar database [15], the analysis of pathogenic and likely pathogenic variations revealed that nonsense mutations account for 17% (Figure 1). Despite the rarity of nonsense mutations in DC, patients carrying this type of genetic variation show very short telomeres with subsequent severe phenotype [58]. Almost 10–20% of DC patients carry *TINF2* mutations. These patients are characterized by early severe manifestations, including bone marrow failure, very short telomeres and dramatically poor life expectancy. Carcinogenesis is more common in DC patients carrying mutations in *TERT* and *TERC* compared to *TINF2*. Unfortunately, *TINF2*-mutated patients generally display poor outcome due to the severity of the disease, regardless of cancer [124]. *RTEL1* mutations account for 5% of DC cases and are associated with heterogeneous clinical manifestations ranging from mild hypocellular bone marrow with B/NK cell lymphopenia to early, very severe cellular deficiency [57]. In addition, other pathological manifestations may be displayed by patients carrying nonsense mutations in *RTEL1* gene such as early onset of thrombocytopenia, anemia, microcephaly, developmental delay, and cerebellar hypoplasia [62].

DC patients are prone to develop solid tumors (typically squamous cell carcinomas of the head, neck and anogenital tract) and hematological malignancies (mostly, non-Hodgkin lymphoma, MDS and AML). At least 4% of DC patients suffer from multiple cancers. According to data reported by Alter and colleagues, the cumulative incidence of all cancers in patients with DC enrolled in National Cancer Institute (NCI) IBMFS cohort is 20% by the age of 50 years. In particular, the risk of MDS and AML are, respectively, increased almost 600 and 73-fold compared with general population [116]. The cumulative risk of AML progression is 10% by age of 70 years in patients who had not received an HSCT. Mortality in non-transplanted patients with DC is mainly due to bone marrow failure, aplastic anemia, infections, and hematological malignancies. Unfortunately, bone marrow transplantation increases the risk of solid tumors because of chronic immunosuppressive therapies [116].

### 2.5. Severe Congenital Neutropenia

Congenital neutropenia (CN) encompasses a variety of inherited bone marrow disorders characterized by the arrest of cellular maturation process within granulocytopoiesis, leading to susceptibility to infections and high risk of leukemic transformation. Congenital neutropenias arise from pathogenic mutations affecting different genes implied in granulocyte differentiation. As a result, granulocyte maturation is generally arrested at promyelocyte stage. Severe neutropenia is defined as an absolute neutrophil granulocyte counts (ANC) less than 0.5 × 10^9^/L (500/μL) [84]. Frequently, other hematological abnormalities are associated with neutropenia, including monocytosis, hypereosinophilia, and polyclonal hypergammaglobulinemia [127].

Severe congenital neutropenia (SCN) was firstly reported in 1956 by Rolf Kostmann, a Swedish physician who analyzed 14 children from an inbreed family, all affected by chronic neutropenia. For that reason, SCN was originally named “Kostmann disease” [128]. Subsequently, several subtypes of congenital neutropenia have been described. Currently, seven different genes have been associated with SCN. Some of these genes leads to autosomal dominant inheritance, such as elastase, neutrophil expressed (*ELANE*) [129], growth factor independent 1 transcriptional repressor (*GFI1*) [130] and T cell immune regulator 1, (*TCIRG1*) [131] and colony stimulating factor 3 receptor (*CSF3R*) for G-CSF [93]. Conversely, HCLS1 associated protein X-1 (*HAX1*), jagunal homolog 1 (*JAGN1*) [132] and glucose-6-phosphatase catalytic subunit 3 (*G6PC3*) [133] segregate with autosomal recessive pattern. Other genes are involved in the pathogenesis of cyclic or intermittent neutropenias, such as solute carrier family 37 member 4 (*SLC37A4*) [134], vacuolar protein sorting 45 homolog (*VPS45*) [135,136], C-X-C motif chemokine receptor 4 (*CXCR4*) [137], *CXCR2* [138] serine/threonine kinase 4 (*STK4*) [139], GATA binding protein 2 (*GATA2*) [140], and WASP actin nucleation promoting factor (*WAS*) [141].

Among all SCN clinical subtypes, SCN1 is the most common, since it has been reported to affect 60–80% of SCN patients. SCN1 is caused by mutations in *ELANE* gene encoding for neutrophil elastase, a cytotoxic serine protease which hydrolyzes multiple protein substrates, including G-CSF receptor, VCAM, c-kit and CXCR4 proteins. According to clinical data collected from the Severe Chronic Neutropenia International Registry, some patients carrying *ELANE* mutations can show cyclic neutropenia with a low risk of evolution to AML, whereas other patients exhibit severe neutropenia with high risk of AML [142,143]. *ELANE*-mutated patients commonly show recurrent fever, skin and oropharyngeal inflammation including ulcers, gingivitis, sinusitis, pharyngitis, and omphalitis which early occur after birth. Untreated children are prone to suffer from diarrhea, pneumonia, and deep abscesses in liver and lungs which are quite common within the first year of life. Nonsense mutations account for 8% of *ELANE* mutations (Table 1) and are often localized inthe fifth or in the final exon with the exception of c.364C>T (p.Q122X) and c.580C>T (p.Q194X), which are identified in the fourth exon. Reasonably, mutations in the final exon should not be a target of NMD since they are likely to yield a stable transcript resulting in translation of a shortened protein (last exon rule). Conversely, a PTC localized in the initial exon is likely to result in the complete loss of protein synthesis. Intriguingly, some patients with SCN harbor digenic mutations in SCN-associated genes. Rare combinations of *ELANE* with *G6PC3* or *HAX1* mutations have been reported, eventually associated with severe neutropenia [144].

The current Kostmann disease (K-SCN) is also recognized as SCN3 and is caused by mutations in *HAX1* gene, encoding HCLS1-associated protein X-1, a mitochondrial protein. Generally, mutations affecting *HAX1* are extremely rare (<1%), even though the prevalence varies significantly in some clustered populations. Germeshausen and colleagues reported that 90% of the patients carrying *HAX1* mutations were of Middle Eastern origin [145]. Mutations in *HAX1* gene have been described in 36% of patients recorded in the Turkish Severe Congenital Neutropenia Registry. In particular, homozygous c.131G>A (p.W44X) nonsense mutation is the most common cause of congenital neutropenia in those populations [146]. Conversely, p.R86X and p.Q190X nonsense mutations are frequently detected in *HAX1*-mutated patients originating from Japan and Sweden [90,145]. In particular, p.Q190X variant is associated with early, severe neutropenia (<0.2 × 10^9^/L) and poor outcome. In addition, Kostmann syndrome is frequently accompanied by mental retardation and epilepsy, especially in patients carrying p.R86X and p.Q190X mutations [127].

*G6PC3* gene encodes for the glucose-6-phosphatase catalytic subunit-3 protein, an essential enzyme involved in gluconeogenesis and glycogenolysis. Mutations in *G6PC3* are associated with SCN4. *G6PC3*-mutated patients suffer from chronic neutropenia, mostly due to an increased apoptosis susceptibility of peripheral neutrophils. This increased apoptotic rate of neutrophil seems to be related to increased endoplasmic reticulum stress induced by loss of *G6PC3*, which usually leads to deficient protein folding within the endoplasmic reticulum [92]. Moreover, SCN4 is characterizedby several non-hematological abnormalities including structural cardiac and urogenital defects, enlarged liver, facial dysmorphism, severe primary pulmonary hypertension, respiratory failure, intermittent thrombocytopenia, and growth and developmental delays. 

Antibiotic administrations, along with G-CSF treatment, are the therapy of choice for SCN. Data collected from the International Severe Chronic Neutropenia Registry, based on a follow-up survey of 3,590 person-years, showed that after 10 years of treatment with G-CSF, the annual risk of MDS/AML was 2.3%. Nevertheless, this risk increases up to 25% after 15 years [147]. In fact, patients who required higher doses of the growth factor due to the lack of response to the treatment, exhibit increased risk of developing myeloid transformation [87].

Similarly to other IBMFS, SCNis considered a pre-leukemic condition. In fact, the rate of hematological malignancy in SCN, regardless of genetic subtype, is far higher than that observed in the general population (10–60% in SCN compared to 1/10,000 inhabitants in the general population) [148].

## 3. The Nonsense Suppression Therapy

The termination of eukaryotic translation process requires the recognition of a stop codon into the A (aminoacyl) site of ribosome by specific aminoacyl-tRNA bounded to eukaryotic translation termination factor 1 (eRF1) and GTP. Rarely, translational mistakes, defined as mispairing, could occur when a near-cognate aminoacyl-tRNA, whose anticodon is complementary just for two of the three nucleotides of a stop codon, improperly binds the stop codon. This process, defined “readthrough”, leads to the incorporation of an amino acid into the nascent polypeptide chain preventing the normal termination of translation. It has been estimated that the endogenous readthrough take place in 0.001% to 0.1% of total tranlsation processes [149] and 0.01% to 1% generally occurs at the PTC [150,151,152]. It follows that PTC may be endogenously inhibited by the natural readthrough leading to a random substitution of the eRF1 with a near-cognate (nc)-tRNA [153,154]. Several factors can affect the readthrough process, including the sequence of nucleotides upstream and downstream the stop codon. It has been observed that the nucleotide which immediately follows the termination codon in the 3′ direction (position +4, considering the first nucleotide of stop codon as +1) is involved in the interactions between mRNA and the translational machinery [155,156,157]. For instance, studies conducted in yeasts have suggested that cytosine at position +4 negatively affect the recognition of eRF1 on the stop codon [158]. Additionally, nucleotides located at positions +5, +6 and +9 can influence the translational readthrough. The relative abundance of various near-cognate aminoacyl-tRNAs is another important aspect [159].

### 3.1. Aminoglycoside Compounds

Aminoglycosides are a class of natural or semisynthetic antibiotics derived from actinomycetes generally used in the treatment of aerobic gram-negative bacilli infections, even though they have also shown antibiotic capabilities against other bacteria including *Staphylococcus sp.* and *Mycobacterium tuberculosis* [160]. Commonly, aminoglycosides share a dibasic aminocyclitol 2-deoxystreptamine (2-DOS) characterized by a core structure of amino sugars connected via glycosidic linkages [161]. Based on components of aminocyclitol moiety and variation of amino and hydroxyl substitutions, different subclasses of aminoglycosides could affect the mechanism of action and susceptibility to various aminoglycoside-modifying enzymes [162,163]. 

The common feature of all aminoglycosides is the bactericidal effect due to the perturbation of peptide elongation at the 30S procariotic ribosomal subunit, which leads to altered protein biosynthesis [161]. Aminoglycosides can bind the A-site on the prokaryotic 16S ribosomal RNA inside the 30S ribosome, modifying its conformation which in turn leads to codon misreading by aminoacyl tRNA and mistranslation [164,165]. Reduced aminoglycoside affinity for the eukaryotic decoding region is due to a key difference in two nucleotides in the eukaryotic ribosomal rRNA sequence compared with prokaryotic rRNA. This affinity for procariotic ribosome allows their clinical use as antibiotics. However, aminoglycosides can partially target the translational machinery of eukaryotic cells.They may induce toxicity affecting mitochondrial translational system, which is similar to the prokaryotic one. 

In eukaryotic cells, aminoglycosides may promote the binding of a near-cognate tRNA to a PTC, displacing class 1 releasing factor, therefore resulting in nonsense codon suppression. PTC avoids the normal translation of human transcripts, except for selenoprotein genes. In those cases, a UGA stop codon can sometimes be translated as a selenocysteine in ribosomes. This process is driven by a quartenary complex consisting of a specialised selenocysteine tRNAsec, a specific elongation factor, a specific RNA secondary structure named SECIS, and GTP. Interestingly, aminoglycosides have been proposed to promote also SECIS-mediated translation [166]. X-ray crystallography and single-molecule FRET analysis revealed that several aminoglycosides such as G418 and gentamicin can directly interact with the 80S eukaryotic ribosome at multiple sites in the large and small subunits. In particular, the 6’-hydroxyl substitution in ring I plays a key role in the binding of the canonical eukaryotic ribosomal decoding center. Therefore, the chemical structure of each aminoglycoside defines the affinity for ribosome interaction and may influence the PTC readthrough efficiency [167].

The first proof of concept concerningPTC-readthrough inducing was represented by geneticin (G418), initially investigated in cystic fibrosis (CF) cell models harboring nonsense mutated *CFTR* gene [168]. Further clinical studies demonstrated the efficacy of aminoglycosides G418 and gentamicin in restoring a significant amount of functional CFTR and dystrophin proteins in CF and Duchenne muscular dystrophy (DMD), respectively (Table 2). But despite this, almost 50% of CF and much fewer DMD patients exhibited the functional rescue of the protein [149].

Severe adverse effects caused by prolonged treatments with aminoglycosides, including auditory and vestibular toxicities have been reported [169]. These side effects limit the widespread clinical use of aminoglycosides for nonsense suppression therapy, even though the addition of antioxidants including D-methionine and melatonin can mitigate the toxic effects sustained by aminoglycosides. Mechanistically, D-methionine and melatonin were shown to reduce ROS production caused by the administration of aminoglycosides such as gentamicin and tobramycin in vitro [159].

### 3.2. Aminoglycoside Derivatives

Several approaches aimed at modifying the chemical structure of aminoglycosides were carried out in the last two decades (Table 2). G418, gentamicin, neomycin and kanamycin analogs were designed and synthesized to improve nonsense suppression capability and to reduce toxicity, finally increasing their possible therapeutic effect as nonsense mutation suppressors [149,181,210]. Since it has been proposed that nonsense suppression effect and toxicity are separate functions within the aminoglycoside chemical structure, the moieties responsible for the cytoplasmic binding of the molecules were re-designed and improved, whereas the structures responsible for mitochondrial damage were made less detrimental. For instance, paromomycin derivative NB30 [182] led to the subsequent development of second generation of aminoglycoside derivatives, termed NB54 and NB84. These molecules have shown reduced toxicity and improved read-through capability compared with gentamicin, in CF, DMD lysosomal storage disease, mucopolysaccharidosis I-Hurler, Rett syndrome and Usher syndrome in vitro and in vivo models [171,176,180,182,183].

Another aminglycoside derivative of neomycin, pyranmycin (TC007) has been identified as a potential PTC-readthrough inducer compound for spinal muscular atrophy (SMA). TC007 restored the full-length survival motor neuron (SMN) protein in human fibroblasts from patients affected by SMA [211]. Moreover, TC007 injection into the central nervous system in a murine model of SMA resulted in longer survival of the motor neurons and increased lifespan of mice [187,188].

Mechanistically, it has been reported that 80S eukaryotic ribosomes contain multiple sites for the binding of paromomycin and TC007 within the ribosomal decoding center [167], similar to other classical aminoglycosides such as G418 and gentamicin.

Recently, the synthetic eukaryotic ribosome-selective glycoside ELX-02 6’-(R)-methyl-5-O-(5-amino-5,6-dideoxyα-L-talofuranosyl)-paromaminesulfate, also known as NB124, has been reported as promising PTC-readthrough inducer compound [185]. ELX-02 can restore the expression of nonsense mutated *CFTR* in CF models by interfering with the NMD process and/or stabilizing the mRNA. ELX-02 exerts its nonsense suppressor activity with improved efficacy and a 100-fold lower antibiotic activity compared with first generation aminoglycosides [186,212]. ELX-02 showed a ten-fold improved PTC-readthrough efficacy compared with gentamicin in restoring nonsense mutated *CFTR* gene in vitro and in vivo [213]. A clinical safety study was conducted in healthy human subjects using increasing doses (from 0.3 to 7.5 mg/kg) of ELX-02 establishing its low renal toxicity and ototoxicity, supporting a further optimization of ELX-02 for therapeutic applications [185]. Importantly, two Phase II clinical trials for ELX-02 in patients with cystic fibrosis (NCT04135495) and nephrophatic cystinosis (NCT04069260) are currently ongoing.

### 3.3. Readthrough Enhancer Molecules

A new class of compounds has been reported to enhance PTC-readthrough sustained by aminoglycosides and their derivatives. These enhancer molecules have been found by high-throughput screening of a huge library of chemical compounds. Five novel compounds, termed CDX3, CDX4, CDX5, CDX10, and CDX11 (Table 2), emerged as enhancers of the nonsense suppressor activity in combination with aminoglycosides, although the use of these compounds alone showed very poor readthrough efficiency [202]. The most promising compound was CDX5. Itincreased up to 180-fold the readthrough activity sustained by G418 at all three PTC sequences (UGA, UAA, UAG). The administration of G418 in combination with CDX5 in patients bearing nonsense mutated tripeptidyl-peptidase-1 (TPP1), dystrophin and SWI/SNF-related matrix-associated actin-dependent regulator of chromatin subfamily A-like protein 1 (SMARCAL1) genes efficiently induced PTC readthrough, justifying the hypothesis of a clinical development of the combination of PTC readtrough inducers plus enhancers [202]. 

Poly-L-aspartic acid, a polyanion with protective effects against alterations induced by aminoglycosides in cultured human kidney proximal tubule cells, showed synergic effect with aminoglycosides. This molecule significanlty increased (up to 40%) the readthrough effect of gentamicin in a murine model of CF [204].

### 3.4. Ataluren and Analogues

In 2007 a new small molecule drug with readthrough activity of premature codons without antibiotic propriety, termed ataluren (also known as PTC124) was launched by PTC Therapeutics [6]. Ataluren, 3-(5-(2-fluorophenyl)-1,2,4-oxadiazol-3-yl)-benzoic acid, is structurally different from aminoglycosides [195]. Compared with classical aminoglycosides, ataluren may promote a more selective readthrough of PTC without affecting endogenous stop codons. Furthermore, ataluren has shown less toxicity and better safety tolerability than aminoglycosides [212,214]. This last feature arises from the fact that ataluren displays its therapeutic activity at concentration much lower (i.e., 3 µM) than gentamicin (i.e., 1 mM) and other aminoglycosides [5,178,197]. It has been demonstrated that the readthrough efficacy of ataluren depends on the sequence of the premature termination codon (PTC) (UAA<UAG<UGA) as well as the sequences of the flanking regions of PTC [195,196,215]. Another key aspect of ataluren efficacy is due by the preferential substitution of each single PTC sequence with particular amino acids, which may affect the functional full-length protein. Ataluren preferentially replaces the UAA codon with codons for Gln (46%), Lys (2%), or Tyr (52%), whereas it replaces UAG codons with those for Gln (88%), Tyr (9%), and Lys (3%), and UGA codons with the those for Trp (86%), Cys (9%), or Arg (5%) [196]. Finally, nonsense suppression efficiency may depend on the NMD. The readthrough activity sustained by ataluren was reported in several studies both in vivo and in vitro [189,192,196,197,200,214], but the mechanism of action of ataluren in cells harboring nonsense mutation remains poorly understood [195,216,217], beyond its proposed interaction with mRNA [218]. The use of ataluren as a potential therapeutic agent for genetic disorders has been early proposed for the treatment of DMD and CF [192,193,214].

Several clinical trials were carried out to evaluate the efficacy and the safety of ataluren in CF patients [219]. Phase II clinical studies showed improvement of CFTR protein expression and function without a clear effect on sweat chloride levels [220]. In order to evaluate long-term safety and the efficacy of ataluren, phase III clinical trials for CF patients were conducted [190,193]. Unfortunately, these studies were discontinued due to the lack of a significant improvement of lung function in treated patients, even though a subcohort of patients reported clinical benefits [193]. No life-threatening adverse events nor case fatalities were reported among the 238 patients. Besides its ambiguous efficacy in CF patients, ataluren is still considered one of the most promising PTC-readthrough inducers. Moreover, ataluren has been already conditionally approved for the treatment of DMD in Europe [221]. Ataluren improved full-length protein synthesis for nonsense-mutated dystrophin gene in DMD patients [192]. Phase III clinical trials reported that ataluren-mediated dystrophin expression is associated with delayed progression in DMD [189]. A study conducted on four non-ambulatory children and adolescents with DMD treated with ataluren demonstrated a mildly attenuated clinical course of the disease, even though a partial reduction of the strength of extensor muscles of the fingers was observed [222]. Long-term outcomes of ataluren treatment in DMD patients showed delayed loss of ambulation and prolongation of daily autonomy [199]. As seen elsewhere, ataluren showed a good safety profile, to the extent that ataluren oral administration has been approved in European Union members states also for pediatric DMD patients starting from two years of age [199].

In 2009 PTC Therapeutics announced a Phase IIa clinical trial to assess the safety and the efficacy of ataluren on patients with Hemophilia A (HA) and Hemophilia B (HB) carrying nonsense mutations (NCT00947193). HA and HB are inherited bleeding disorders caused by mutations in the gene for factor VIII (FVIII) and factor IX (FIX), respectively, which are essential for blood clotting. Approximately 10–30% of HA and HB patients carry nonsense mutations associated with severe condition. The primary endpoint of this study was the improvement of plasma FVIII and FIX functional activity. Although the study was terminated in 2011, no data have been published so far.

Another Phase II, crossover study of ataluren (NCT02758626) is ongoing for the treatment of drug-resistent epilepsy caused by Dravet syndrome and cyclin-dependent kinase-like 5 (CDKL5) deficiency. Safety profile of ataluren will be characterized in patients carrying nonsense mutations. Furthermore, changes in convulsive or drop seizure frequency, cognitive, motor, and behavioral functions will be evaluated upon twelve weeks of treatment. However, a recent publication by Landsberger’s group casts doubt on the final effectiveness of that clinical trial, because preclinical results showed that both ataluren and another non-aminoglycoside drug, termed GJ072, failed to induce PTC-readthrough on nonsense mutated *CDKL5* gene in vitro [172].

Of note, almost 50% of patients suffering from aniridia, an inherited disorder causing defect in iris development, optic nerve hypoplasia, cataract, glaucoma, and progressive corneal opacity, exhibit in-frame nonsense mutations. For that reason, ataluren has been proposed as an hopeful therapeutic hypothesis also in this case. More than 600 mutations in paired box (*PAX)-6* gene cause aniridia. Ataluren has been tested in a semi-dominant Pax6^Sey/+^ mouse model of aniridia carrying a nonsense mutation (G194X) [198]. Results indicated that ataluren treatment can improve *PAX6* expression in vivo. In addition, the expression of a downstream effector of *PAX6*, namely the matrix metalloprotease (MMP)-9, which is required for the maintenance and repair of the corneal epithelium, was also increased. Interestingly, since the sequence of Pax6^Sey/+^ mouse PTC is UGA, Wang and colleagues postulated that the normal Gly194 would most likely be substituted with Trp, leading to a tolerated change within the linker region where Gly194 is normally located. Accordingly with these findings, a recent study reported that ataluren can improve *PAX6* expression by inducing 30–40% of translational readthrough in primary lymphocytes isolated from patients with aniridia carrying nonsense mutations [200]. These findings provide rationale for the ongoing Phase II clinical trial aimed at assessing the safety and efficacy of ataluren for the treatment of aniridia caused by nonsense mutations (NCT02647359). The study has been designed to carry out a 144-week treatment with an optional 96-week open label extension sub-study and will soon beterminated. 

In 2010 ataluren has been clinically tested for the treatment of methylmalonic acidemia, an inherited polygenic disease caused by mutations in genes encoding the mitochondrial enzyme methylmalonyl-CoA mutase (*MCM*) or for adenosylcobalamin (*AdoCbl*), also known as coenzyme B12. Loss of expression of *MCM* or *AdoCbl* causes the release of elevated levels of methylmalonic acid (MMacid) in blood, urine, and other tissues. Since approximately 5–20% of patients with mutations in the *MCM* gene, and 20–50% of patients with mutations in *AdoCbl* genes carry nonsense mutations, ataluren was proposed as a possible therapeutic option. A Phase IIa trial (NCT01141075) evaluating the safety and efficacy of ataluren on methylmalonic acidemia was carried in pediatric patients (age 2 years and older), aimed at observing a decrease of MMacid levels in blood and urine. Although the study was concluded in 2012, no results have been published so far. 

In August 2019 a further open label Phase I-II clinical trial investigating the efficacy and the safety of the combination of ataluren with pembrolizumab for the treatment of metastatic mismatch repair deficient and proficient colorectal adenocarcinoma and metastatic mismatch repair deficient endometrial carcinoma was sponsored by the University of Amsterdam (NCT04014530). Pembrolizumab (commercially available as Keytruda^®^) is a clinically approved humanized monoclonal antibody able to block the interaction between the programmed cell death protein (PD)-1 and its ligands which is currently used for the treatment of non-small-cell lung carcinoma (NSCLC) and other solid tumors [223]. Ataluren combination with Pembrolizumab was hypothesized because many deficient metastatic mismatch repair cancers often possessout-of-frame PTCs. PTC readthrough of this code was proposed to generate new target peptides which might be detected by the immune system. This should enhance the effect of pembrolizumab’s anti-PD1 therapy. 

In addition, we recently studied the effect of ataluren on the nonsense mutated *SBDS* gene in different hematological and non-hematological cells obtained from 13 patients with SDS [197]. In those experiments, ataluren restored full-length SBDS protein expression in bone marrow hematopoietic progenitors, mesenchymal stromal cells, and lymphoblasts. SDS bone marrow failure is mainly dominated by defective myeloid differentiation in bone marrow precursor cells. Restoration of SBDS protein synthesis was associated with a significant improvement of myeloid differentiation as determined by hematopoietic colony assays. We had reported that SDS is characterized by hyper-phosphorylation of the mammalian target of rapamycin (mTOR) and signal transducer and activator of transcription (STAT)-3 proteins [224]. Several reports showed that SDS hematopoietic cells exhibit increased apoptosis rate [225,226]. Interestingly, ataluren reduced both mTOR and apoptotic rate in SDS cells [197]. More than half of SDS patients exhibit a unique nonsense mutation, namely the c.183-184-TA>CT, unlike other inherited diseases, including other IBMFS. Thus, our cohort of patients was genetically homogeneous with the same stop codon, UGA, and possibly avoiding fluctuation of ataluren efficacy due to different PTC sequences. Interestingly, this stop codon has the same sequence reported for the Pax6^Sey/+^ aniridia mouse model discussed above, and similar positive preclinical results in full-length protein expression were obtained. Despite this, almost 23% of ex vivo experiments showed no effect upon ataluren treatment, sometimes in terms of restoration of SBDS protein synthesis, in some cases in terms of functional effect (myeloid differentiation). Another limitation of this study was that no other functional aspects related to restored *SBDS* expression, such as decreased *TP53* expression and STAT3 phosphorylation, was tested due to the paucity of primary cells collected from bone marrow aspirates. Further analyses should be carried out in order to clarify these additional evidence. Finally, the assessment of the effect of ataluren on other tissues affected by SDS, such as bones (e.g. osteoblasts, chondrocytes), pancreatic epithelial cells and cells of the nervous system would be helpful to better clarify ataluren preclinical efficacy. These observations provide a rationale for ataluren as a potential treatment for SDS.

However, some critical limitations emerged, as stated above. Ataluren can compete with other aminoglycoside antibiotics, therefore reducing its efficacy. These observations were already reported in the clinical studies conducted on patients with CF, who receive aminoglycoside antibiotics such as tobramycin [193]. To obtain new compounds with improved efficacy compared to ataluren, Pibiri’s research group has designed and synthesized novel derivatives of ataluren through the modification of the oxadiozole heterocyclic core or its lateral portion. Pibiri and colleages modified the aromatic ring of ataluren allowing a different electron distribution within the molecule, which leads to an improved interaction with lipids, including the biological membrane, therefore influencing the absorption of new derivatives and ameliorating their pharmacokinetic [227,228]. These ataluren analogues displayed a reduced cytotoxicity in vitro, compared to G418. It has been hypothesized that these new compounds may not share the same biological target of aminoglycosides [227]. In particular, the 1,3,4-oxadiazole termed NV2445 showed promising results in terms of readthrough efficacy in vitro, proving to be more effective than ataluren [201]. However, further studies on these molecules are needed to clarify the possible mechanism of action and the toxicology in vivo.

### 3.5. NMD Inhibitors

The success of nonsense suppression therapy depends on both the readthrough activity of a specific compound and the nonsense mediated decay which often take place in response to nonsense mutations and can limit the efficiency of the PTC-readthrough sustained by nonsense suppressor molecules. NMD is an evolutionarily conserved process that surveys newly synthesized mRNAs and degrades those that present a PTC. Since truncated polypeptides can damage the normal cellular functions, eukaryotic cells have developed a NMD, a defense mechanism by which the mutated transcripts containing a PTC are rapidly subject to degradation [4,229]. 

NMD takes place when the PTC is recognized by the protein complex consisting of UPF1, UPF2, UPF3, suppressor of morphogenetic effect on genitalia 1 (SMG1), SMG8, SMG9, DEAH box polypeptide 34 (DHX34), and the Exon–Exon Junction protein Complex. UPF2 binds to UPF1 amino-terminal domain, causing the release of this domain from the UPF1 central core. Subsequently, UPF1 is phosphorylated by SMG1, leading to mRNA degradation due to the recruitment of proteins which trigger endonucleolytic cleavage, normally sustained by SMG6, or by deadenylation and decapping [229]. However, sometimes PTC may not trigger NMD. For instance, PTC located at less than 50 nucleotides upstream of the last exon–exon junction typically do not trigger NMD (this is also known as the 50nt rule) as well as PTCs in the last exon of a gene also do not trigger NMD (last exon rule). Moreover, PTCs located 150 nucleotides downstream the start codon typically fail to trigger NMD (the start-proximal rule), probably because of translation re-initiation [229]. 

Interestingly, some inhibitors of UPF1 including the serine/threonine protein kinase SMG1, wortmannin and caffeine can inhibit NMD in vitro. Studies conducted in *Caenorhabditis elegans* revealed that SMG1 mediates in vivo NMD by phosphorylating UPF1. SMG1 is one of the components of the so-called SURF complex, which is composed of other SMGs kinases, UPF1 and eukaryotic release factors [230]. It has recently been reported that antisense oligonucleotides designed to interfere with SMG1 can inhibit NMD restoring CFTR protein synthesis in a model of cystic fibrosis carrying the W1282X nonsense mutation in vitro [231]. Moreover, knockdown of SMG-8, a subunit of the SMG-1 complex, restored collagen type VI α 2 mRNA and protein expression in Ullrich congenital muscular dystrophy fibroblasts carrying homozygous frameshift mutation generating a PTC [232]. Similarly results were found knocking down SMG-8 in a cerebral autosomal recessive arteriopathy model obtained from a patient who carried nonsense mutations in HtrA serine peptidase 1 gene [232]. 

The methylxanthine alkaloid, caffeine, and the covalent inhibitor of phosphoinositide 3-kinases (PI3K), wortmannin, represent other UPF1 inhibitors that can reduce NMD, as reported in Ullrich congenital muscular dystrophy models [209]. Both caffeine and wortmannin can inhibit UPF1 phosphorylation through the inhibition of the SMG1 kinase [233,234].

Caffeine prevented NMD in HEK293 cells expressing a construct plasmid encoding for the R577X nonsense mutated *ACTN3* gene, involved in skeletal muscle fast fiber contraction [179]. Furthermore, caffeine-mediated NMD inhibition was shown to restore c.715C>T nonsense mutated *CHM* mRNA expression near to healthy control levels in a model of x-linked recessive chorioretinal dystrophy, Choroideremia [208]. Interestingly, NMD inhibition sustained by caffeine enhanced ataluren-mediated readthrough in a nonsense mutated cell model of cystic fibrosis, strengthening the hypothesis of a combinatory therapy of NMD inhibitors and readthrough inducers [207].

One particular case concerning NMD inhibitors is represented by the anti-inflammatory drug amlexanox (commercialized as Aphthasol^®^), already approved for the treatment of aphthous ulcers. In 2012 amlexanox emerged from a tethering-screening system of a large chemical library as a putative NMD inhibitor [205]. In addition to its UPF-1-mediated NMD inhibitor capability [206], amlexanox surprisingly produced PTC-readthrough in cell models of cystic fibrosis and Duchenne muscular dystrophy. More recently, these data were confirmed in cells obtained by patients affected by recessive dystrophic epidermolysis bullosa, where amlexanox successfully restored nonsense mutated Collagen type VIIα1 chain (COL7A1) gene both in terms of mRNA and full length protein [206].

### 3.6. Modified t-RNA

Suppressor (sup)-tRNA are modified aminoacylated-tRNA that compete with translation termination factors, forcing the incorporation of an amino acid in place of PTC. The advantage of sup-tRNA consists of the specific amino acid substitution, avoiding the introduction of missense mutations due to the randomness of classical readthrough. Sup-tRNA have been already tested for the restoration of several nonsense mutated genes in cell models of β-thalassemia [235], xeroderma pigmentosum [236] and Ulrich disease [237]. Although sup-tRNA have shown high stop codon specificity and higher readthrough capability compared with other nonsense mutation suppressor molecules, sup-tRNA have shown also several limitations, including very low in vivo efficacy due to poor cellular uptake [238]. More recently, a new generation of anticodon engineered (ACE)-tRNA displayed higher PTC-suppression potency both in vitro and in vivo, reporting the restoration of expression of multiple genes including *CFTR* [239].

## 4. Discussion and Perspectives 

No pharmacological therapies designed specifically against an IBMFS have been developed yet. The application of gene therapy and induced pluripotent stem cells (iPS) sounds promising. At least two lentiviral gene-based trials are recruiting for the treatment of Fanconi anemia (NCT01331018 and NCT03157804). In addition, gene editing has recently raised interest of IBMFS community. The correction of *FANCA*, *FANCC*, *FANCD1*/*BRCA2*, *FANCI*, and *FANCF* pathogenic variants by CRISPR-Cas9 gene editing has been demonstrated in vitro [240,241,242,243]. Gene editing approach based on CRISPR/Cas9-sgRNA has also been recently shownto reduce *ELANE* expression ex vivo in bone marrow hematopoietic progenitors from patients with SCN. CRISPR-Cas9-mediated knockdown of *ELANE* significantly induced neutrophil maturation in vitro [244]. It should be nevertheless noted that CRISPR-Cas9 technology presents several limitations for its rapid translation as a therapy, including cell type dependent delivery, incomplete efficiency of homologous recombination and the possibility of off-target editing [245].

Because 28% of FA, 24% of SCN, 21% of DBA, 20% of SDS, and 17% of DC mutated alleles (Figure 1) carry nonsense mutations in IBMFS-related genes, we propose another approach, that of nonsense suppressor therapy. Nonsense suppression therapy used in non-hematologic genetic diseases such as DMD and CF. Ataluren (Translarna^®^) is an approved drug for the treatment of DMD and, importantly, several Phase II/III clinical studies reported very low toxicity of ataluren even in pediatric patients aged two and older [189,190,191,192,193,194,219].

Little is known about their effect on IBMFS. We recently reported that ataluren improves SBDS full-length protein synthesis and function in bone marrow hematopoietic progenitors and mesenchymal stromal cells isolated from bone marrow biopsies of patients with SDS [197]. While ataluren has not yet be tested for other IBMFS, we anticipate a clinical study in SDS.

PTC-readthrough inducer molecules generally exhibit low efficiency. Although ataluren showed encouraging preclinical results in CF models, restoring CFTR protein synthesis and chloride function, the clinical development has been subsequently discontinued because of poor clinical benefits in terms of respiratory function improvement. A post hoc subgroup analysis demonstrated that a sub-cohort of patients treated with ataluren reported some clinical benefit. A recent clinical study showed that a partial synergistic effect of ivacaftor and ataluren can be observed in terms of improvement of nasal potential difference, although the major limitation of this study was the very little number (only two) of patients tested [191].

Ataluren however failed in a model of Dravet syndrome, an autosomal dominant form of epilepsy, perhaps because of codon selectivity. Its efficacy may depend on the sequence of PTC (UAA<UAG<UGA) [196,215]. Interestingly, both aniridia and SDS models, where ataluren significantly improved the target protein synthesis and function with promising in vivo and ex vivo results, shared the same stop codon, namely UGA, which is the hypothetical best sequence. Alternatively, readthrough efficacy may depend on the tissue and cell type targeted by the therapy [217]. Positive responses from blood cells derived from both aniridia and SDS justify the use of ataluren in other IBMFS.

Besides ataluren, several other readthrough inducer molecules have been synthesized and preclinically tested so far. Some ataluren analogues have shown increased in vitro efficacy compared to ataluren [201,227]. However, little is known about the toxicity of these molecules and further studies are needed to clarify this important step in drug development.

Another strategy aimed at restoring nonsense mutated transcripts is due by NMD inhibitors. However, as previously discussed, since the endogenous readthrough occurs very infrequently, a therapeutic approach aimed only at inhibiting the NMD, without increasing the readthrough capability, might not be sufficient for clinical purposes. One strategy would be to combine the two approaches to enhance full-length protein synthesis from transcripts of nonsense-mutated genes. For example, PTC-readthrough enhancers may potentiate the effect of reathrough-inducer drugs.

L-leucine administration improved the anemiain *rps19*-deficient zebrafish model of DBA by activating the mTOR pathway [246]. Moreover, L-leucine treatment has been proposed to activate translation of erythroid cells, improving globin gene synthesis and ameliorating the anemic phenotype in *rps19* and *rpl11* mutants in zebrafish [106]. These studies support a Phase I/II clinical trial (NCT01362595) that evaluates the effect of L-leucine supplementation on red blood cell transfusion dependent DBA patients. Since L-leucine may promote protein synthesis in erythroid cells, combinationtherapy of L-leucine, NMD inhibitors and/or PTC-readthrough inducers could improve the anemia of DBA.

All these premises suggest therefore that nonsense suppression therapy should be tested in IBMFS with hopeful expectations. Even if only preclinical positive results have been achieved so far, they should be considered as new important proof of concepts for extending the current therapeutic scenario for IBMFS.

## 5. Patents

VB and MC are the inventors of the patent for the specific use of ataluren in SDS, WO2018050706-A1, entitled “Method of treatment of Shwachman–Diamond syndrome”.

## Figures and Tables

**Figure 1 ijms-21-04672-f001:**
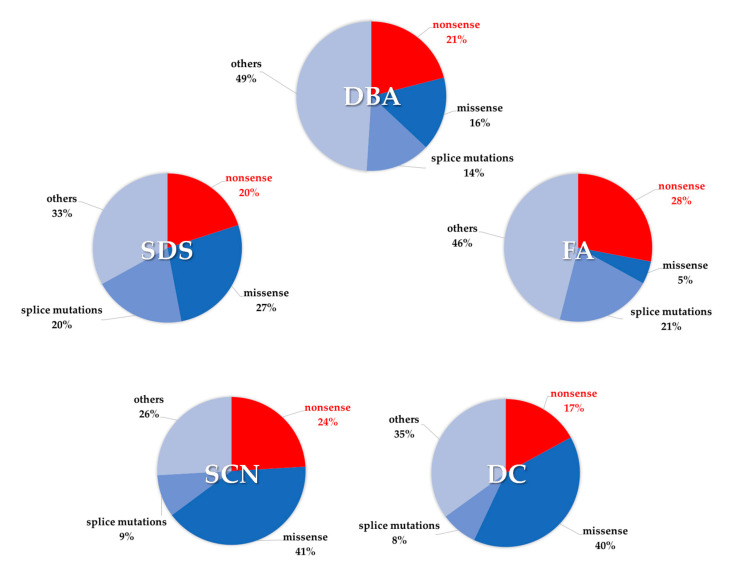
Incidence of nonsense mutations in inherited bone marrow failure syndromes (IBMFS). Percentages have been calculated on the basis of ClinVar database [15]. Only pathogenic and likely pathogenic variants have been taken into account.

**Table 1 ijms-21-04672-t001:** IBMFS-associated genes mostly affected by nonsense mutations.

Disease	Gene	Incidence of Nonsense Mutations	Gene Function	Ref
Diamond–Blackfan anemia	RPL5 RPL11 RPL35A RPS10 RPS17 RPS19 RPS24 RPS26	33% 7% 22% 17% 25% 20% 25% 20%	Pre-rRNA processing of the 18 rRNA, formation of the 40S or 60S ribosome subunit	[33,34,35,36,37,38,39,40,41,42,43,44,45,46,47,48,49,50,51,52,53]
Shwachman–Diamond syndrome	SBDS	20%	60S ribosome assembly	[23,24,25,54]
Dyskeratosis Congenita	CTC1 PARN RTEL1 TERT TINF2	28% 21% 26% 27% 18%	Telomere protection Poly(A)-specific ribonuclease Stability and elongation of telomeres Telomerase reverse transcriptase Stability and elongation of telomeres	[55,56,57,58,59,60,61,62]
Fanconi anemia	FANCA FANCB FANCC FANCD1 FANCE FANCF FANCG FANCI FANCJ FANCL FANCN	14% 16% 33% 59% 43% 25% 24% 29% 30% 14% 37%	DNA cross-link repair, chromosome stability	[33,37,63,64,65,66,67,68,69,70,71,72,73,74,75,76,77,78,79,80,81,82,83]
Severe congenital neutropenia	CSF3R ELANE G6PC3 HAX1 JAGN1	40% 8% 33% 29% 14%	G-CSF receptor Neutrophil elastase Hydrolysis of glucose 6-phosphate Apoptosis control, cytoskeletal development, myeloid regulator Neutrophil differentiation and survival	[33,84,85,86,87,88,89,90,91,92,93,94,95,96]

Data from ClinVar database [15]. Percentages have been calculated taking into account only pathogenic and likely pathogenic genetic variations.

**Table 2 ijms-21-04672-t002:** State-of-the-art of the nonsense suppression therapy.

Molecule	Function	Clinical Trials	Ref
G418 (geneticin)	Readthrough inducer	None	[152,168,170,171,172]
Gentamicin	Readthrough inducer	Phase II in CF (NCT00376428)	[168,171,173,174,175,176,177,178,179]
NB30	Readthrough inducer	None	[176,180]
NB54	Readthrough inducer	None	[171,176,180,181,182,183]
NB84	Readthrough inducer	None	[176,184]
ELX-02	Readthrough inducer	Phase II in NephrophaticCystinosis (NCT04069260) Phase II in CF (NCT04126473)	[185,186]
Pyramycin (TC007)	Readthrough inducer	None	[187,188]
Ataluren (PTC124)	Readthrough inducer	Phase IIa in HA and HB (NCT00947193) Phase IIa in methylmalonic academia (NCT01141075) Phase II in Dravet syndrome (NCT02758626) Phase II in aniridia (NCT02647359) Phase II in CF (NCT00351078; NCT00237380; NCT00458341; NCT00234663) Phase II in DMD (NCT00264888) Phase III in CF (NCT02139306; NCT02107859; NCT02456103) Phase III in DMD(NCT02456103; NCT02139306; NCT00803205; NCT01140451; NCT02107859) Phase IV in CF (NCT03256968; NCT03256799)	[189,190,191,192,193,194,195,196,197,198,199,200]
NV2445	Readthrough inducer	None	[201]
CDX3, CDX4, CDX5, CDX10, CDX11	Readthrough enhancer	None	[202,203]
Poly-L-aspartic acid	Readthrough enhancer	None	[204]
Amlexanox	NMD inhibitor/ Readthrough inducer	None	[205,206]
Caffeine	NMD inhibitor	None	[179,207,208]
Wortmannin	NMD inhibitor	None	[209]

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
