# Peer review of "Nonsense Suppression Therapy: New Hypothesis for the Treatment of Inherited Bone Marrow Failure Syndromes"

_ijms, 2020, doi:10.3390/ijms21134672_

Round 1
Reviewer 1 Report
The manuscript from Bezzerri et al. is a very interesting and up to date review. It is well written. The first part with the description of each diseases is quite long and the reader that will be interested in this review to understand why nonsense suppression therapy could apply to BMFS might feel loss. The first part should be reduced with a short description for the diseases and keep all information related with nonsense mutations.
Beside of this comment, I listed some points that should be addressed to help the reading of this review:
- page 2 line 65: authors wrote "These mutations produce premature termination codons (PTC), which lead to the production of truncated, nonfunctional proteins." but that is not correct since no truncated proteins are found when a PTC is present on an mRNA due to the activation of NMD.
- Amlexanox is described as NMD inhibitor in table 2 but it was reported as NMD inhibitor and PTC readthrough activator as indicated further in the review.
- Line 544, change UAA into UAG. In the same line, the percentage for Tyrosine is wrong (95%), it is more 8-9%. The percentage for Lysine is also wrong since it is written 35% when it is between 3 and 4%.
- Line 561, the following sentence is ambiguous:” Phase III clinical trials reported that ataluren-mediated dystrophin expression was associated with improvement of dystrophin protein function in patients, as measured by six minutes walk distance test” since it gives the impression that the physical condition of patients improves when they actually have a reduce loss of physical strength.
- Minor points: Line 520, a space is missing for “It increase
Line 528, a space is missing.
Line 692/ “UPf1” should be written “UPF1”
Author Response
The manuscript from Bezzerri et al. is a very interesting and up to date review. It is well written. The first part with the description of each diseases is quite long and the reader that will be interested in this review to understand why nonsense suppression therapy could apply to BMFS might feel loss. The first part should be reduced with a short description for the diseases and keep all information related with nonsense mutations.
R1.1.We thank the Reviewer for the kind appreciation of our work. We have slightly reduced the sections concerning IBMFS as suggested.
Beside of this comment, I listed some points that should be addressed to help the reading of this review:
page 2 line 65: authors wrote "These mutations produce premature termination codons (PTC), which lead to the production of truncated, nonfunctional proteins." but that is not correct since no truncated proteins are found when a PTC is present on an mRNA due to the activation of NMD.
R1.2. We modified the text accordingly (line 65).
Amlexanox is described as NMD inhibitor in table 2 but it was reported as NMD inhibitor and PTC readthrough activator as indicated further in the review.
R1.3. We corrected the Table 2 accordingly.
Line 544, change UAA into UAG. In the same line, the percentage for Tyrosine is wrong (95%), it is more 8-9%. The percentage for Lysine is also wrong since it is written 35% when it is between 3 and 4%.
R1.4. We apologize for this inconvenience. Probably decimals were improperly removed throughout internal review. We have now fixed all discrepancies. We thank the reviewer for pointing out this issue (lines 508-510).
Line 561, the following sentence is ambiguous:” Phase III clinical trials reported that ataluren-mediated dystrophin expression was associated with improvement of dystrophin protein function in patients, as measured by six minutes walk distance test” since it gives the impression that the physical condition of patients improves when they actually have a reduce loss of physical strength.
R1.5. We agree with the Reviewer. We have reworded the sentence which sounded confounding. We also added the clinical study in which loss of physical strength was observed. Finally, we extended the discussion including a new clinical study based on STRIDE registry (lines 526-532).
Minor points: Line 520, a space is missing for “It increase
Line 528, a space is missing.
Line 692/ “UPf1” should be written “UPF1”
R1.6. We corrected these typos.

Reviewer 2 Report
This review article describes the inherited bone marrow failure syndromes (IBMFSs), which are caused by the disruption of proteins involved in DNA repair, telomere biology, and ribosome biogenesis. Approximately 20% of patients show nonsense mutations in the IBMFS-related genes. Therefore, the authors considered the therapeutic potential of nonsense suppression therapy for IBMFS. This is indeed an important contribution to the field of IBMFS therapy. However, please note and address the following points for improvement and clarifications:
- In order to enhance readers’ understanding, illustrations for the following sections should be strengthened: 1) the mechanisms of premature termination codons (PTC)-readthrough by aminoglycoside compounds and 2) the nonsense-mediated decay (NMD) of complex molecules and the mechanisms of NMD inhibition by Amlexanox, Caffeine, and Wortmannin.
- Please elaborate on the mechanism through which CXD5 increases PTC-readthrough activity.
- On page 14 line 544, “it replaces UAA codons with…” should be “it replaces UAG codons with…”.
- Missing spaces between words or punctuations were noticed as follows:
Page 2 line 53: “correctbone”
Page 3 line 117: “in 1964[17]”
Page 6 line 234: “genes[118].”
Page 7 line 239: “abnormalities[109]”
Page 7 line 276: “variants[70]”
Page 9 line 367: “andglucose-6-phosphatase”
Page 9 line 381: “birth.Untreated”
Page 10 line 414: “infections , sever”
Page 11 line 469: “mutatedCFTR gene”
Page 13 line 520: “Itincreased”
Page 13 line 528: “aminoglycosides.This”
Page 15 line 617: “possessout-of-frame”
Page 16 line 649: “whoreceive”
Page 17 line 724: “recruitingfor”
Page 18 line 727: “demonstratedin vitro”
Page 18 line 740: “reportedthat”
Page 18 line 768: “ofnonsense-mutated genes”
- A punctuation was missing on page 16 line 655. Please add a period after “[233,234]”.
Author Response
This review article describes the inherited bone marrow failure syndromes (IBMFSs), which are caused by the disruption of proteins involved in DNA repair, telomere biology, and ribosome biogenesis. Approximately 20% of patients show nonsense mutations in the IBMFS-related genes. Therefore, the authors considered the therapeutic potential of nonsense suppression therapy for IBMFS. This is indeed an important contribution to the field of IBMFS therapy.
R2.1: We thank this Reviewer for the kind appreciation of our work
However, please note and address the following points for improvement and clarifications:
In order to enhance readers’ understanding, illustrations for the following sections should be strengthened: 1) the mechanisms of premature termination codons (PTC)-readthrough by aminoglycoside compounds and 2) the nonsense-mediated decay (NMD) of complex molecules and the mechanisms of NMD inhibition by Amlexanox, Caffeine, and Wortmannin.
R2.2: We thank this reviewer for the useful suggestion. Accordingly, we have improved the explanation of mechanism of action of PTC-readthrough sustained by aminoglycosides (lines 419-430), aminoglycoside derivatives (lines 463-465), ataluren (lines 513-514) and the mechanisms of NMD inhibition (lines 638-644), wortmannin and caffeine (lines 664-665) and amlexanox (line 677).
Please elaborate on the mechanism through which CXD5 increases PTC-readthrough activity.
R2.3: ELX-02 readthrough mechanisms has been elaborated accordingly (lines 468-470).
On page 14 line 544, “it replaces UAA codons with…” should be “it replaces UAG codons with…”.
R2.4. We apologize for this typo. Text has been now reworded (line 509).
Missing spaces between words or punctuations were noticed as follows:
Page 2 line 53: “correctbone”
Page 3 line 117: “in 1964[17]”
Page 6 line 234: “genes[118].”
Page 7 line 239: “abnormalities[109]”
Page 7 line 276: “variants[70]”
Page 9 line 367: “andglucose-6-phosphatase”
Page 9 line 381: “birth.Untreated”
Page 10 line 414: “infections , sever”
Page 11 line 469: “mutatedCFTR gene”
Page 13 line 520: “Itincreased”
Page 13 line 528: “aminoglycosides.This”
Page 15 line 617: “possessout-of-frame”
Page 16 line 649: “whoreceive”
Page 17 line 724: “recruitingfor”
Page 18 line 727: “demonstratedin vitro”
Page 18 line 740: “reportedthat”
Page 18 line 768: “ofnonsense-mutated genes”
A punctuation was missing on page 16 line 655. Please add a period after “[233,234]”.
R2.5: Text has been corrected accordingly.
